# New Diagnostics for Fungal Infections in Transplant Infectious Disease: A Systematic Review

**DOI:** 10.3390/jof11010048

**Published:** 2025-01-09

**Authors:** Apurva Akkad, Neha Nanda

**Affiliations:** Division of Infectious Disease, Department of Medicine, Keck School of Medicine, University of Southern California, 1500 San Pablo St., Los Angeles, CA 90033, USA

**Keywords:** fungal diagnostics, transplant infectious disease, molecular diagnostics, metagenomic sequencing, magnetic resonance, gas chromatography

## Abstract

Fungal infections are common in highly immunosuppressed, solid organ transplant recipients. They can be quite difficult to diagnose in a timely manner; thus, we present a review of current studies focusing on broad categories of molecular diagnostics, i.e., metagenomic sequencing, magnetic resonance, and gas chromatography mass spectrometry. We further discuss their syndrome-specific utilization in the diagnosis of fungemia and disseminated disease, pneumonia, and central nervous system infections. We assess the level of evidence of their utility as fungal diagnostics particularly in solid organ transplant recipients using the STARD criteria. In addition, we provide future research directions to substantiate and appropriately utilize these platforms in clinical practice. Directed polymerase chain reaction testing and targeted metagenomic sequencing are being used clinically and show the most promise, though only in conjunction with conventional methods at this time. The majority of these platforms contain limited data, and thus further larger studies are needed in order to properly implement their use.

## 1. Introduction

Solid organ transplantation (SOT) has increasingly become a sought-after treatment for patients with end-organ disease. Life expectancy after SOT has improved with newer immunosuppressive therapies and with improved peri- and post transplantation care. There is an estimated 3.4 million life years that have been saved over the past decade with SOT [1]. However, with longer life expectancy after SOT [2], infection in solid organ transplant recipients (SOTRs) is one of the leading causes of post-transplantation morbidity [3]. Solid organ transplant recipients (SOTRs) are at a heightened risk for infection due to impaired immune responses. While conventional diagnostic methods like culture, histopathology, serology, antigen detection, and molecular testing continue to play a central role in the diagnosis of infection, they are often limited by suboptimal sensitivity and prolonged turnaround times. This contributes to missed or delayed diagnoses in SOTRs [4]. Over the past ten years, there has been a rapid emergence of novel diagnostic platforms. These platforms detect a broad range of pathogens with faster turnaround times [5], with the additional marketed benefit of increased sensitivity [6]. However, evidence that such platforms have a beneficial impact on infection diagnosis in SOTRs is still in the nascent stages.

Fungal infections are responsible for greater than 1.5 million deaths globally per year, with a significant proportion in immunocompromised patients [7]. There is abundant literature regarding novel diagnostic platforms for bacterial and viral diseases; however, there is a relative dearth of evidence regarding their use for fungal disease. Diagnostic platforms for bacterial and viral pathogens have been well studied for bloodstream infections, disseminated infection, pneumonia, and neurological infections. These large studies have yielded variable results.

In this review, we aim to assess the utility of novel platforms to diagnose fungal infections in the SOTR population. It is not well known how often fungal infections are missed due to the relative difficulty in their diagnosis; however, it is suspected that, as their prevalence is increasing, more are prone to being overlooked. We will specifically examine molecular studies, metagenomic sequencing, magnetic resonance, gas chromatography mass spectrometry, and the level of evidence for their utility as fungal diagnostics.

Molecular studies including multiplex polymerase chain reaction (PCR) matrix-assisted laser desorption ionization (MALDI), fluorescence in situ hybridization (FISH) [8], as well as 18s rRNA are becoming more widely used for fungal diagnostics. These assays detect an array of pathogens in order to provide a broader range of results with a more rapid turnaround time. Metagenomic next-generation sequencing (mNGS) is a culture-independent, hypothesis-independent, broad-spectrum sequencing method. It typically uses DNA sequencing, which excludes RNA-based pathogens (viruses). These platforms typically use cell-free DNA (cfDNA) or next-generation sequencing (NGS). A drawback is that there is no common standard for result interpretation [9]. Magnetic resonance, or the T2MR platform, is a PCR technology combined with magnetic resonance to detect superparamagnetic nanoparticles bound to amplicon-specific probes. It detects DNA from pathogens directly from blood. Currently, this technology has leveraged T2 bacteria and T2 candida panels and has received approval from the Food and Drug Administration (FDA) for the diagnosis of bloodstream infections [10]. Gas chromatography has not been utilized in clinical settings but has potential for use in the diagnosis of pulmonary invasive fungal infections. This platform utilizes the detection of exhaled volatile organic compounds produced during fungal metabolism [11].

In this review, we evaluate the utility of the aforementioned platforms for fungal infection diagnosis in SOTRs. To provide a clinical context, we assess these platforms under the following clinical syndromes: (a) disseminated disease; (b) pneumonia; (c) central nervous system disease.

## 2. Results

### 2.1. Fungemia and Disseminated Disease

In our search, we identified two studies with strong evidence for the utility of molecular platforms as diagnostics: one study with strong evidence and another with moderate evidence for metagenomic platforms (based on the STARD score), and two studies with strong evidence for the T2MR (magnetic resonance technology) candida assay. In this section, we provide details of select studies that showed strong and moderate evidence, as well as highlight two molecular studies with weak evidence. We outlined salient diagnostic assays for fungemia and disseminated disease in Table 1.

(A)Molecular Platforms

For the diagnosis of fungemia and disseminated disease, the ePlex molecular assay and T2MR platforms are the most promising, followed by the BioFire FilmArray (PCR) and Karius platforms (metagenomic next-generation sequencing—discussed below).

The ePlex multiplex PCR panel (Genmark Diagnostics), a molecular assay, found 100% concordance for candidemia, with a positive predictive value (PPV) of 99% among all isolates [14]. The study had a small sample size for fungemia (7), though there was sufficient power for bacterial isolates (210 bacterial isolates). No baseline characteristics or eligibility criteria were provided; however, strong evidence was noted for the use of the ePlex panel in the proper clinical setting. This study received a Standards for Reporting of Diagnostic Accuracy (STARD) score of 27/30 (strong evidence).

Among the nucleic acid-based tests, the Sepsitest (SeptiFast), which targets the ribosomal DNA of fungi, was found to be 50% sensitive and 86% specific, with a PPV of 26.3%, among fungal bloodstream infections, thus showing significant limitations in its clinical utility [16]. This study received a STARD score of 28/30 (strong evidence).

There was weak to moderate evidence for the utility of the BioFire FilmArray (molecular assay), which is an FDA-approved multiplex polymerase chain reaction (PCR) panel for use on positive blood cultures. Rule et al. evaluated its clinical utility in the intensive care unit (ICU) setting, for the modification of empiric antimicrobial therapy in critically ill patients with sepsis. The assay provided results to clinicians significantly earlier than conventional culture methods. This, in turn, allowed for a modification of empirical antimicrobial therapy to a more appropriate therapy in 25/78 (32%) patients. Additionally, they found that the use of the BioFire FilmArray BCID panel permitted a prompt implementation of additional infection prevention and control practices in a sizeable proportion (14%) of the patients in the study who harbored multidrug-resistant pathogens [12]. This study received a STARD score of 20/30 (weak evidence). The study had high contamination rates (29.4%) and a small sample size (78) and did not provide the baseline characteristics of the patients and the sensitivity, specificity, or accuracy values. Caméléna et al. [13] conducted a multicenter evaluation of the FilmArray panel (molecular), with a mean positive percent agreement between the BioFire FilmArray assay and standard culture of 97% for all pathogens. The agreement ranged from 67% for *Candida albicans* to 100% for 17 targets included in the BCID2 panel [2]. This study received a STARD score of 24/30 (moderate), as the authors did not provide the baseline characteristics of the patients included, eligibility criteria, or distribution of the severity of illness. Thus, there is weak to moderate evidence of the clinical utility of the BioFire FilmArray panel, and it may be used as an adjunct to standard culture and diagnostic methods.

The Sepsis Flow chip assay, a molecular assay, showed 95% diagnostic accuracy overall, with 93.3% sensitivity and 100% specificity (including a high number of bacterial isolates); however, only five yeast samples were included [15]. The study received a STARD score of 19/30, as it had a small sample size (five), no baseline patient characteristics were shown, and there was no comparison to conventional methods.

The Fungiplex candida PCR, as well as the Lightcycler SeptiFast test was compared to conventional blood cultures. This assay had a sensitivity of 100%, a specificity of 94.1%, and a 94.6% diagnostic accuracy. The SeptiFast, in comparison, had a sensitivity of 60% and a specificity of 96.1% [17]. This study received a STARD score of 26/30, as no true reference standard was established, it did not mention the distribution of the severity of illness, had inadequate power, and generalizability was limited because the authors included patients in the intensive care unit only. Thus, there is moderate evidence for the Fungiplex assay, and further prospective studies are needed to indicate its use for fungemia diagnosis.

(A)Metagenomic Next-Generation Sequencing

Metagenomic sequencing (mNGS) of plasma cell-free DNA has emerged as a potential new diagnostic modality allowing for broad pathogen detection with a rapid turnaround time. The turn-around time can depend on where the test is performed and on specific institutional protocols. Such assays include those of Karius and iDtect. The Karius assay, as per Hogan et al., in a retrospective cohort study of 82 patients, demonstrated that the results led to a positive impact in 6/82 (7.3%) patients, a negative impact in 3/82 (3.7%) patients, and no impact in 71/82 (86.6%) patients. It was indeterminate in 2/82 (2.4%) patients [18]. This study received a STARD score of 29/30, demonstrating strong evidence for the cautious use of the Karius test, as the vast majority of the patients had no impact, and 4% of the cases had a negative impact.

Untargeted next-generation sequencing (uNGS), such as with the iDtect assay, in a proof-of-concept study, noted that untargeted next-generation sequencing had a high negative predictive value compared with conventional methods (64/65, 95% CI 0.95-1) [19]. However, there was significant detection of potentially clinically irrelevant pathogens. It is unclear from the study if these pathogens represented colonization or were true pathogens that did not require therapy. A STARD score of 25/30 was given, as the study omitted the baseline characteristics of the included patients, had a small fungal sample size, and did not mention the distribution of alternative diagnoses. This study provided moderate evidence for the potential use of the uNGS assay. Additional studies are needed before this technology is adopted in clinical practice.

(B)Magnetic Resonance

Magnetic resonance with the T2MR candida panel demonstrated specificity and sensitivity of 99.4% and 91.1% (compared to blood cultures, 50%), respectively, with high negative predictive values (99–99.5%) across a wide range of pre-test probabilities. However, its limitations include the inability to detect some clinically important *Candida* species (*C. guilliermondii*, *C. lusitanea*, *C. kefyr*, and *C. auris*), which account for 1–10% of *Candida* infections in transplant patients [20]. A STARD score of 28/30 was given, showing strong evidence for the future use with this panel. Furthermore, another study with T2 showed that this assay was more likely to yield a positive result than blood cultures in patients receiving antifungal medications, with sensitivity and specificity of 89% and 98%, respectively [21], suggesting that at least some of the T2-positive/blood culture-negative discordant results may represent infections that were missed by standard cultures. A STARD score of 29/30 was given to this study, demonstrating strong evidence regarding the potential of the T2MR candida panel for fungemia detection.

### 2.2. Pneumonia

Of the platforms that we evaluated, the Qiagen and Roche (molecular platforms) are the most promising, followed by mNGS and WGS. In our search, we identified four studies with strong evidence for the utility of molecular platforms and mNGS, along with one study with moderate evidence for the utility of gas chromatography. In this section, we provide details of select studies that reported moderate and strong evidence for the utility of some diagnostic methods, as well as one highlighting weak evidence for an mNGS platform. The reference standard for these studies typically was conventional current therapy, which includes culture and histopathology. We outlined the salient diagnostic assays for pneumonia in Table 2.

(A)Molecular Platforms

For fungal pneumonia, most molecular platforms, such as Luminex (NxTag), Nanosphere (Verigene), FilmArray (Biofire), eSensor (GenMark), and Curetis (Unyvero), do not detect fungal organisms in their current configuration. Cuenca-Estrella et al. studied the Qiagen fungal PCR panel, which showed 91.6% sensitivity and 94.4% specificity in patients with febrile neutropenia for *Aspergillus* spp. This assay can also detect pneumocystis with a sensitivity of 85–100% across multiple studies [22]. The STARD score for this study was 28/30, with strong evidence for *Aspergillus* spp. detection application with rapid turnaround times of 24–48 h (compared to both CT imaging and galactomannan testing). Roche developed an *Aspergillus* spp. PCR assay that was found to have 62% sensitivity, 100% specificity, 100% PPV, and 71% negative predictive value (NPV) for *Aspergillus* spp. detection in liver transplant patients [23]. This assay was only tested in patients in whom the galactomannan assay was positive; thus, its sensitivity was low, but it was shown to have high specificity and PPV. The STARD score for this study was 28/30, indicating strong evidence of future benefit of this assay, when utilized in appropriate clinical settings.

(B)Metagenomic Next-Generation Sequencing

Hilt et al. conducted a study using IDbyDNA explify (Illumina) in immunosuppressed pediatric patients. The results of mNGS were concordant with those of standard microbiologic testing for bacteria (90.2%) and viruses (94.1%) but not fungi (66.7%). In 18 of 41 (44%) children, mNGS identified possible pathogens missed by conventional testing [24]. The STARD score for this study was 21/30, suggesting weak evidence of the utility of this assay for fungal infections. Further studies need to be performed in adults to determine its future potential. Whole-genome sequencing (WGS) was studied on lung biopsy specimens, showing 57.1% sensitivity and 61.5% specificity for fungi. The positive predictive value (PPV, 44.4% for fungi) was much lower than the negative predictive value (NPV, 72.7% for fungi) for WGS vs. the culture method. WGS showed the highest specificity (100.0%) and PPV (100.0%) in the evaluation of fungi, when compared with the histopathology method. In all the five stem cell transplant patients with confirmed fungal pneumonia on lung biopsy (by histopathology and/or culture), mNGS identified fungal organisms [25]. Based on the STARD criteria, this study was given a score of 28/30, suggesting strong evidence of the utility of NGS in select circumstances, when used in conjunction with culture and histopathology. Wu et al. conducted a multicenter, randomized controlled trial for patients with severe community-acquired pneumonia (CAP). In an intention-to-treat analysis, the time to clinical improvement was better in the mNGS group than in the conventional medical therapy group (10 d vs. 13 d) due to more rapid pathogen detection. The proportion of patients with clinical improvement within 14 days was significantly higher in the mNGS group (62.0%) than in the CMT (conventional medical therapy) group (46.5%) [26]. Notably, 18% of the isolates detected by mNGS contained fungi as the primary pathogen, and 5% showed a fungal co-infection. In addition, 10% of the isolates detected by CMT contained fungi as the primary pathogen, with 7% of the isolates with a fungal co-infection. When mNGS was combined with CMTs, the time to clinical improvement for patients with CAP was reduced. The STARD scoring was 28/30 (strong evidence). This study did show strong evidence for the utility of mNGS in severely ill patients, in conjunction with conventional diagnostics.

(C)Magnetic Resonance

Magnetic resonance, or the T2MR candida panel, was described above (in fungemia), as its primary role is in the diagnosis of candidemia, with no studies in fungal pneumonia.

(D)Gas chromatography

Koo et al. ran a proof-of-concept study, finding 94.0% sensitivity and 93.0% specificity for invasive pulmonary aspergillosis (IPA) in patients with suspicion of fungal pneumonia [27]. This study was given a STARD rating of 26/30, suggesting moderate evidence and potential of clinical utility of this technique in the future. Gas chromatography is also being tested for mucor, the assay currently being in development. Gas chromatography is not yet in commercial use; however, its primary potential use will be in fungal pneumonia.

### 2.3. Central Nervous System (CNS) Infections

For diagnosing CNS infections, the metagenomic next-generation sequencing (mNGS) platform seems to be the most promising, followed by the BioFire FilmArray, a molecular platform. In our search, we identified one study with strong evidence for the utility of mNGS and two studies with moderate evidence for the utility of molecular PCR. In this section, we provide details of select studies that had moderate and strong evidence. We outlined salient diagnostic assays for CNS infections in Table 3.

(A)Molecular Platforms

For central nervous system (CNS) infections, the BioFire FilmArray, a molecular panel, has been studied in a few trials. Leber et al. showed that the FilmArray ME Panel demonstrated a sensitivity or positive percentage of agreement of 100% for 9 of 14 analytes, with 1500+ CSF specimens analyzed. The specificity or negative percentage of agreement was 99.2% or greater for all other analytes, including *cryptococcus neoformans* and *gattii* [28]. This study received a STARD score of 27/30. It showed strong evidence of potential utility, with further studies needed for additional fungal pathogens. In another retrospective observational study for cryptococcal meningitis, the FilmArray ME panel had a sensitivity of 71.4% and a specificity of 100% [29] for cryptococcal meningitis. Thus, this showed that one cannot rule out a diagnosis with a negative test. This study received a STARD score of 26/30, i.e., showed moderate evidence of utility of the method used. Thus, this assay may only be used as an adjunctive test until further studies are conducted to assess its clinical utility.

(B)Metagenomic Next-Generation Sequencing

Metagenomic sequencing in CNS infections was studied in a multicenter study of 58 patients with confirmed meningoencephalitis (including stem cell transplant or solid organ transplant recipients); 19 (33%) were diagnosed by conventional testing and mNGS, 26 (45%) by conventional testing alone, and 13 (22%) by mNGS alone [30]. This study was given a STARD score of 29/30, showing strong evidence of the future potential of mNGS in the diagnosis of CNS infections, with current evidence supporting the use of both conventional testing and mNGS. The available evidence does support the possible use of mNGS alone in select circumstances or in the case of an elusive diagnosis, as a substantial number of patients were diagnosed with mNGS alone. Larger studies on CNS fungal infections need to be conducted to evaluate the role of mNGS in clinical practice.

## 3. Discussion

We evaluated the utility of new diagnostic assays in diagnosing fungal infections, specifically, fungemia, pneumonia, and CNS infections. This is an emerging area of diagnostics whose clinical utility and circumstances for use is not yet well established. At this time, the clinical utility of newer diagnostic assays is dependent on clinician’s discretion and is often decided on a case-by-case basis. Further studies are needed across various pathogens, but particularly in the diagnosis of fungal infections in immunosuppressed individuals.

Regarding fungemia, assays such as the BioFire FilmArray or the mNGS Karius test have shown to have utility in clinical practice and have received FDA approval. However, the evidence for the clinical use of these assays alone, i.e., without the concomitant use of conventional assays, is not strong. Strong evidence for the cautious use of mNGS with tests such as with the Karius test has been shown. Result interpretation needs to be performed by an expert in the field, and its utility varies by clinical scenario. It has become apparent that microbial DNA is ubiquitous; therefore, there may be reagent contamination, poor or incomplete primers, or simply a less-than-optimal-quality databank that may account for mNGS assays not showing good data to support monotherapy [31]. Assays that currently have good evidence of utility and may have potential clinical utility in the future, include the ePlex assay, the Fungiplex assay, uNGS (iDtect), as well as magnetic resonance, with the T2MR candida panel. Lastly, there is strong evidence against the use of the Sepsitest, a molecular assay, in clinical practice, today.

For fungal pneumonia, at this time, there are no assays that have been shown to have utility in clinical practice. Assays that may have potential clinical utility in the future include molecular tests such as those of Qiagen and Roche, which have been shown in studies to have strong evidence of utility supporting their use in select circumstances, which thereby limits their generalizability. mNGS showed both weak evidence of utility with IDbyDNA and strong evidence with other NGS platforms in conjunction with conventional methods, such as culture and histopathology. Gas chromatography may have potential for future use in fungal pneumonia based on the moderate evidence of its utility available at this time.

Regarding CNS infections, assays such as the BioFire FilmArray meningoencephalitis panel have some utility in clinical use, particularly for cryptococcal meningitis, in addition to adjunctive diagnostic methods. mNGS was found to have strong evidence of utility for future use in the diagnosis of CNS infections, with current evidence supporting the use of conventional testing and mNGS together. Further studies specifically on CNS fungal infections will need to be carried out due to the relatively low number of fungal infections identified in this patient population in the scientific literature.

The limitations of this review include the limited sample size for fungal isolates in the reviewed studies. Most of the studies were retrospective or observational in design, which led to inherent limitations. In select studies, the reference standard for evaluating the assay used was not clearly stated, which limited the interpretation of the results. There is a need for large, prospective trials to further our understanding of fungal diagnostics in SOTRs.

Fungal infections, whether disseminated or localized, can be challenging to diagnose, especially in SOTRs. Such novel, rapid diagnostic assays have an impact on both the short term and the long-term outcomes in the peri-transplantation and post-transplantation period. In the short term, these assays will be instrumental for a timely diagnosis and the prompt initiation of anti-fungal therapy and discontinuation of anti-bacterials. In the long term, these novel assays will be pivotal to retain the gut microbiome in SOTRs, thereby reducing the mortality in this vulnerable population [32].

## 4. Conclusions

Molecular methods, mNGS, magnetic resonance, and gas chromatography provide four new exciting platforms that have the potential for accurate detection of fungal isolates with rapid turnaround times, in a non-invasive manner. These newer platforms do show promise; however, further studies are needed to determine their validity, especially if they are to be used as standalone tests. Molecular methods and mNGS are currently being used, in conjunction with conventional methods. However, the active involvement of infectious disease expertise is imperative to ensure stewardship and an accurate interpretation of the test results. Magnetic resonance and gas chromatography are not yet as far along but have potential for use in clinical care in the future.

## 5. Materials and Methods

We searched PubMed for relevant studies that were conducted using the novel diagnostic panels with inclusion of fungal organisms. We used the following search terms, along with the assay to be evaluated: fungemia, fungal pneumonia, and central nervous system (CNS) infections. We used the STARD criteria to evaluate the available scientific literature [33]. To stratify the available evidence, we classified the studies in three categories—strong evidence, moderate evidence, and weak evidence. We agreed on the following stratification: a score of 27 or higher constituted strong evidence, a score of 22–26 corresponded to moderate evidence, and a score of 21 or lower indicated a study with weak evidence. The reference standard for all platforms was conventional diagnostics, which included cultures or histopathology, unless otherwise specified.

## Figures and Tables

**Table 1 jof-11-00048-t001:** Fungemia and disseminated disease: studies of various platforms with scoring, level of evidence, and relevant information.

Study	Test	Technology	STARD	Level of Evidence	Notes
Rule et al., 2021 [12]	Biofire Filmarray	Molecular	20	Weak	Modification of empirical antimicrobial therapy to more appropriate agents in 32% of patients High contamination rates (29.4%), a small sample size (78), did not provide sensitivity, specificity, or accuracy values
Caméléna F, Péan et al., 2022 [13]	Biofire Filmarray	Molecular	24	Moderate	Mean positive agreement between BCID and standard culture 97%, no distribution of severity of illness, no baseline characteristics
Huang et al., 2019 [14]	ePlex	Molecular	27	Strong	Positive predictive value (PPV) of 99% among all isolates Small sample size for fungemia (7), though otherwise sufficient power for bacterial isolates (210 bacterial isolates)
Galiana et al., 2017 [15]	Sepsis Flow Chip	Molecular	19	Weak	93.3% sensitivity and 100% specificity, only 5 yeast samples
Warhurst et al., 2015 [16]	Sepsitest	Molecular	28	Strong	50% sensitivity, 86% specificity, and 26.3% PPV
Fuchs et al., 2019 [17]	Fungiplex	Molecular	26	Moderate	Fungiplex: sensitivity of 100%, specificity of 94.1%, and a 94.6% diagnostic accuracy, SeptiFast: sensitivity of 60% and a specificity of 96.1, low power—pilot study
Hogan et al., 2021 [18]	Karius (mNGS)	Metagenomic	29	Strong	Retrospective cohort study: positive impact in 7.3% of patients, a negative impact in 3.7% of patients, and no impact in 86.6%
Parize et al., 2017 [19]	iDtect (mNGS)	Metagenomic	25	Moderate	High negative predictive value compared with conventional methods (64/65), significant detection of potentially clinically irrelevant pathogens
Mylonakis et al., 2015 [20]	T2MR	Magnetic Resonance	28	Strong	Specificity and sensitivity of 99.4% and 91.1%, negative predictive values (99–99.5%), limitations include an inability to detect some clinically important Candida species (*C. guilliermondii*, *C. lusitanae*, *C. kefyr* and *C. auris*),
Clancy et al., 2018 [21]	T2MR	Magnetic Resonance	29	Strong	Sensitivity 89%, Specificity 98%

**Table 2 jof-11-00048-t002:** Fungal pneumonia: studies of various platforms with scoring, level of evidence, and relevant information.

Study	Test	Technology	STARD	Level of Evidence	Notes
Cuenca-Estrella, 2009 [22]	Qiagen (molecular)	Molecular	28	Strong	*Aspergillus* spp.: 91.6% sensitivity, 94.4% specificity in patients with febrile neutropenia. *Pneumocystis* sensitivity 85–100%
Botterel et al., 2008 [23]	Roche (molecular)	Molecular	28	Strong	62% sensitivity, 100% specificity, 100% PPV, 71% negative predictive value. Only tested in galactomannan positive patients
Hilt et al., 2022 [24]	IDbyDNA (NGS)	Metagenomic	21	Weak	Fungi 66.7% concordance with conventional testing (very low)
Li et al., 2018 [25]	WGS	Metagenomic	28	Strong	57.1% sensitivity, 61.5% specificity for fungi. The positive predictive value (PPV) 44.4% for fungi was much lower than negative predictive value (NPV) (72.7% for fungi) in mNGS vs. culture method. mNGS showed the highest specificity (100.0%) and PPV (100.0%) in the evaluation of fungi, when compared with histopathology
Wu et al., 2024 [26]	NGS	Metagenomic	28	Strong	Patients with clinical improvement within 14 days mNGS group (62.0%), CMT (conventional medical therapy) group (46.5%)
Koo et al., 2018 [27]	gas chromatography	Gas Chromatography	26	Moderate	94.0% sensitivity and 93.0% specificity for invasive pulmonary *Aspergillus* spp., proof of concept study

**Table 3 jof-11-00048-t003:** Central nervous system disease: studies of various platforms with scoring, level of evidence, and relevant information.

Study	Test	Technology	STARD	Level of Evidence	Notes
Leber et al., 2016 [28]	Biofire Filmarray	Molecular	27	Strong	Positive percentage of agreement 100% for 9 of 14 analytes, 1500+ CSF specimens analyzed. The specificity, or negative percentage of agreement 99.2% or greater for all other analytes
Walker et al., 2018 [29]	Biofire Filmarray	Molecular	26	Moderate	Sensitivity 71.4% Specificity 100% for cryptococcal meningitis
Wilson et al., 2019 [30]	mNGS	mNGS	29	Strong	58 pts confirmed ME, 19 (33%) were diagnosed by conventional testing and mNGS, 26 (45%) by conventional testing alone, and 13 (22%) by mNGS alone

## Data Availability

No new data were created or analyzed in this study. Data sharing is not applicable to this article.

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
