# Peer review of "New Diagnostics for Fungal Infections in Transplant Infectious Disease: A Systematic Review"

_jof, 2025, doi:10.3390/jof11010048_

Round 1

Reviewer 1 Report

Dear Authors;

The theme presented in the manuscript submitted to JoF is of great relevance for academic studies and also for current medical mycology, which is constantly neglected by health professionals, to the detriment of other microbial groups: bacteria and viruses.

The text is clear and well written. However, I suggest adjustments to the tables as specified below:

The format of the tables is not appropriate. The concept of a table is different from that presented by the authors. The table consists of only 3 lines. Please correct it. In addition, the font and quality of the table are poor, which makes it difficult to read. There are nomenclature errors, genus and species of microorganisms are written in italics and capital (Aspergillus spp), not "aspergilllus" or "Aspergillus". Please correct it.

1. Correct all tables.

2. Correct spelling of fungus species: lines 214, 216, 218, 219 (Aspergillus spp.), table 3

3. et al in the tables: Correct in italic form. Add year in references in all the tables.

Reviewer 2 Report

^Thank you for allowing me to review this manuscript.  It covers in good detail the major modalities of diagnostics with clinically relevant comments.  As is noted above, the introduction might focus a bit more on the topic at hand (fungi) - and place this into context - how often do we miss important isolates/diagnoses? Or is this unknown? 

Tables are useful and contain a lot of good information.  Not sure the STARD score table adds to this paper. 

Some minor comments:

"one study with strong evidence and another with moderate evidence for metagenomic 110" - might mention that this is based on STARD Score (it is in methods but useful here). 

"with a rapid turna- 169 round time" - this may depend on where this is performed and communications barriers. 

"However, there was significant detection of potentially clinically irrelevant pathogens. 180" - it is unclear what these detected species mean.  How many of these were fungal?  If the patients have colitis or other focal infections, could they be real but not require therapy? 

"B) Metagenomic Next Generation Sequencing 224" - is it worth considering the reasons why NGS may be less good than anticipated? Nucleic acid extraction? Sequence databank? Sample handling? Poor primers? others? In other words, future directions? 

"Wu et al. conducted a multicenter, randomized con- 240 trolled trial for patients with severe community acquired pneumonia (CAP). In the inten- 241 tion-to-treat analysis, the time to clinical improvement was beÄ´er in the mNGS group than 242 that in the conventional medical therapy group (10 d vs. 13 d) due to more rapid pathogen 243 detection. The proportion of patients with clinical improvement within 14 days was sig- 244 nificantly higher in the mNGS group (62.0%) than that in the CMT (conventional medical 245 therapy) group (46.5%).xxvi When mNGS is combined with CMTs time to clinical improve- 246 ment for patients with CAP was reduced. The STARD scoring was 28/30 (strong evidence). 247 This study did show strong evidence for utility of mNGS in severely ill patients, in con- 248 junction with conventional diagnostics." - this is all NON-fungal I believe?  Is this relevant? 
